# Acaricidal activity of Egyptian crude plant extracts against *Haemaphysalis longicornis* ticks

**Ahmed M. Abdou**[1,2], **Nanang R. Arifeta**[1], **Abdel-latif S. Seddek**[2], **Samy Abdel-Raouf Fahim Morad**[3], **Noha Abdelmageed**[4], **Mohamed O. Badry**[5], **Rika Umemiya-Shirafuji**[1], **Yoshifumi Nishikawa**[1]*

1 National Research Center for Protozoan Diseases, Obihiro University of Agriculture and Veterinary Medicine, Obihiro, Hokkaido, Japan, 2 Department of Forensic Medicine and Toxicology, Faculty of Veterinary Medicine, South Valley University, Qena, Egypt, 3 Department of Pharmacology, Faculty of Veterinary Medicine, South Valley University, Qena, Egypt, 4 Department of Pharmacology, Faculty of Veterinary Medicine, Sohag University, Sohag, Egypt, 5 Department of Botany and Microbiology, Faculty of Science, South Valley University, Qena, Egypt

* nisikawa@obihiro.ac.jp

**Data Availability Statement:** The data presented in this study was available in the main article text and its supplementary information files.

## Abstract

*Haemaphysalis longicornis* is a common Ixodida tick species found in temperate areas of Asian countries. An anti-tick assay was conducted on adult female *H. longicornis* ticks. Plant extract solutions were prepared at concentrations of 50, 25, and 10 mg/mL. Tick survival and mortality were assessed by counting the number of dead and live ticks at 24 h, 48 h, 72 h, and 96 h posttreatment. Out of 11 plant extracts screened, *Artemisia judaica* extract exhibited the highest potency with 100% mortality (5/5) at 48 h when applied at high and moderate concentrations (50 and 25 mg/mL). Similar results were observed at 96 h for the 10 mg/mL group compared to the untreated ticks. *Cleome droserifolia* extract demonstrated partial activity with 60% (3/5) and 20% (1/5) mortality at 96 h posttreatment at concentrations of 50 and 25 mg/mL, respectively. *Forsskaolea tenacissima* extract showed a weak effect with 100% tick mortality (5/5) only at the highest treatment concentration after 96 h. To confirm the activity of *A. judaica*, trial 2 was conducted. *A. judaica* demonstrated potency within 48 h in high dose and 72 h in moderate dose, with 100% mortality (15/15) at 96 h posttreatment compared to untreated ticks. The median lethal time 50 ($LT_{50}$) values were 30.37 h for the high and 55.08 h for the moderate doses. Liquid chromatography–mass spectrometry was performed on the most potent candidate (*A. judaica*) to identify its phytochemical components. The results revealed the presence of 9 compounds identified through manual annotation and 74 compounds from the Global Natural Products Social library. These compounds included terpenoids, steroids, phenylpropanoids, flavonoid glycosides, flavonoids, and benzenoids. Camphor was identified as the major component via both approaches. These findings suggest the potential use of *A. judaica* extract in the future development of acaricidal therapeutics.

**Funding:** This study was supported by KAKENHI Grants from the Japan Society for the Promotion of Science (20F20402 [Y.N.]).

**Competing interests:** The authors have declared that no competing interests exist.

## Introduction

*Haemaphysalis longicornis* (Acari: Ixodidae), commonly known as a bush tick or New Zealand cattle tick, is endemic to the East Asia/Pacific region and is found in Japan, Korea, eastern China, southeast Russia, Australia, New Zealand, and some Pacific islands [1]. However, the geographic range of this tick has increased, and it has recently been confirmed in the USA [2]. The tick has been found feeding on humans, domestic animals (dogs, cats, cattle, sheep, goats, and horses), and a variety of species of wildlife in the USA [2]. *H. longicornis* can transmit multiple pathogens, such as *Anaplasma* spp., *Rickettsia* spp., *Babesia* spp., and severe fever with thrombocytopenia syndrome virus (SFTSV) [3].

To minimize the adverse clinical effects of *H. longicornis* infestation and reduce the risk of transmission of canine and zoonotic pathogens, effective treatments are necessary [1]. Fluralaner, administered orally at a dose of 25–50 mg/kg, provides up to 114 days of protection against *H. longicornis* tick infestations in dogs [4]. Sarolaner (Simparica®) has shown efficacy in treating *H. longicornis* nymph infestations and preventing reinfestation for up to 5 weeks posttreatment without adverse effects when given as a single oral dose of 2 mg/kg in dogs [1]. In the agricultural sector, tick-infested livestock can be protected using pour-on or dip formulations, employing methods such as the vacuum cleaning approach in grazing cattle [5, 6] or by selecting tick-resistant host breeds [7]. In the pet industry, topical acaricides are commonly used for treatment [2, 8]. Another successful approach is spraying grazing pasture with amitraz, although this method is less cost-effective than topical acaricides unless performed in small areas with high tick densities [9]. In a previous study, an *in vitro* feeding assay system using fipronil and ivermectin was established, whereas survival of *Ixodes ricinus* adult female ticks' survival was monitored daily over 9 days through a silicone membrane on bovine blood treated with different doses of fipronil and ivermectin ranges from 0.001 to 10 μg/mL [10].

The main strategy for controlling tick infestations has been high-cost chemical treatment. However, the emergence of resistant tick strains poses a significant challenge [11]. A summary of previous studies conducted worldwide, focusing on the resistance of ticks to different commercial acaricides applied to infected livestock, particularly cattle, between 1992 and 2020, was provided by Dzemo et al., 2022 [12]. One potential source of alternative acaricidal treatments could be natural resources, such as plants. Therefore, this study aimed to evaluate the effectiveness of some crude Egyptian plant extracts against adult female ticks of *H. longicornis*.

## Materials and methods

### Ethics statement

This study was performed in strict accordance with the recommendations of the Guide for the Care and Use of Laboratory Animals of the Ministry of Education, Culture, Sports, Science and Technology, Japan. The protocol was approved by the Committee on the Ethics of Animal Experiments at Obihiro University of Agriculture and Veterinary Medicine, Obihiro, Japan (permit numbers 19–74).

### Plant materials collected from the desert

The plants used in this study were obtained through a field survey conducted along two desert roads in the southern region of Egypt near the Qena governorate (latitude: 26˚ 09' 51.05" N, longitude: 32˚ 43' 36.16" E). The collection took place at two specific sites on the Qena-Sohag and Qena-Safaga desert roads in Qena, and 1 candidate was collected from the Luxor governorate, Egypt. The collection activities were carried out in accordance with the permission, ethics, and guidelines provided by the Faculty of Veterinary Medicine, South Valley

University, Qena. For reference, a map of the collection sites is provided in S1 Fig. During the middle of May 2019, the plant taxa were collected, and the coordinates of the plant collection sites were previously presented in our prior study [13]. These collected plant samples were identified using established studies [14–18] and cross-referenced with the herbarium at the Faculty of Science, South Valley University, Qena, Egypt. Additionally, an official letter of identification was obtained to validate the plant taxonomy and species. Furthermore, the taxonomy and species information were updated based on the information available from Plants of the World Online [19].

## Plant material preparation and extraction

The plant samples were dried in the shade for 3–10 days. Once dried, the leaves, flowers, fruits, and seeds were processed into a fine powder using a kitchen blender. Subsequently, 100 g of each powdered plant material was dissolved in either 80% methanol, 70% ethanol, or distilled water. The extraction process took place over 1–3 days, with a ratio of 100 g of plant powder to 1 L of the chosen solvent. After the addition of the solvent, the macerated or solid plant parts settled at the bottom of the vessel underneath the liquid plant supernatant. The plant supernatant was collected and filtered into a wide conical flask using a glass filtration apparatus. The solution was then air-dried in a wide Petri dish at room temperature for 1–3 days. The resulting crude extract was collected in centrifuge tubes and stored at -30˚C until further use.

To evaluate the acaricidal efficacy of the plant extracts, they were individually solubilized in 80% methanol to prepare stock solutions at concentrations ranging from 100 to 10 mg/mL. The methods for plant extraction were conducted following previously reported studies [13, 20]. A summary of the previously reported medicinal uses and the Latin binomial names of the wild plants used in this study were previously provided [13, 20]. The Latin binomial names of the plants, as well as their respective families, are listed in S1 Table.

## Anti-tick assay of adult female *H. longicornis* ticks

The assay was conducted using adult female *H. longicornis* ticks (OKAYAMA strain). Plant extract solutions, along with the positive control drug and negative control, were prepared at the concentrations listed in Table 1. These solutions were evenly spread onto pieces of filter paper (diameter = 26 mm), each of which was then placed in a Petri dish (diameter = 30 mm). To remove the solvent, the Petri dishes were left to dry at room temperature for 72 h. After the drying process, adult female ticks were transferred to the Petri dishes, with each concentration of extract from each plant candidate having its own Petri dish. In total, 270 ticks were used in our study, 180 ticks were used for the experiment (Trial 1), with five ticks placed in each Petri dish per each plant concentration. While, in trial 2, 90 ticks were used for the experiment with fifteen ticks being used per plant concentration. The Petri dishes were then incubated at a temperature of 25±1˚C and a relative humidity >80%. To determine the survival and mortality rates, the number of dead and live ticks was recorded. The survival and mortality rates were assessed at specific time intervals, i.e., 24 h, 48 h, 72 h, and 96 h.

## Liquid chromatography with mass spectrometry (LC–MS) analysis for molecular networking

The crude extract was dissolved in a mixture of water and methanol at a ratio of 25:75, resulting in a concentration of 1.0 mg/mL for LC–MS analysis. To prepare the sample for analysis, the stock solution was further diluted with water to achieve a final concentration of 0.1 mg/L. The analysis was performed using a high-resolution mass spectrometer, Q Exactive, which was connected to a high-performance liquid chromatography Ultimate 3000 RSLC (Thermo Fisher

**Table 1. Preparation of plant material for the anti-tick assay.**

| Plant name | Type of plant extraction | Solvent | Highest concentration in stock (mg/mL) | Working concentrations (mg/mL) | The final volume added/filter paper (µL) |
|---|---|---|---|---|---|
| *Artemisia judaica L.* | Methanolic | 80%-MeOH) | 100 | 50 | 500 |
| *Cleome droserifolia (Forssk.) Delile* | | | | 25 | 250 |
| | | | | 10 | 100 |
| *Trichodesma africanum (L.) Sm.* | | | | | |
| *Ochradenus baccatus Delile* | | | | | |
| *Forsskaolea tenacissima L.* | | | | | |
| *Anabasis setifera Moq.* | | | | | |
| *Aerva javanica (Burm.f.) Juss. ex Schult.* | | | | | |
| *Carthamus tinctorius L.* | | | | | |
| *Ocimum basilicum L.* | Ethanolic | | | | |
| *Citrullus colocynthis (L.) Schrad.* | | | | | |
| *Origanum majorana L.* | Aqueous | | | | |
| **Positive control (Cypermethrin)** | | | 10 | 5 | 500 |
| | | | | 1 | 100 |
| **Negative control (80% MeOH)** | | | 0 | | 500 |

Scientific, United States) equipped with an InertSustain AQ-C18 (2.1 × 150 mm; 3 µm particle, GL Science, Japan). During the analysis, elution was carried out using a mobile phase consisting of $H_2O$ + 0.1% formic acid (A) and acetonitrile (B), which were pumped at a rate of 0.2 ml/min. The gradient program was set as follows: 2% B (0–3 min), 2–98% B (3–30 min), 98% B (30–35 min), 98–2% B (35–35.1 min), and 2% B (35.1–40 min. The column oven was set at 40°C, and the injection volume was 2 µL. LC–MS/MS analyses were achieved by coupling the LC system to an Orbitrap MS (Q ExactiveTM, Thermo Fisher Scientific, United States).

### MS/MS-molecular networking-based dereplication

The raw data obtained from the LC–MS/MS system were converted to mzXML format using the ProteoWizard tool (Vanderbilt University, United States) [21]. For further analysis, the MZmine workflow for feature-based molecular networking on GNPS was used. The molecular network was generated using the online workflow available on the GNPS website (http://gnps.ucsd.edu) [22]. To preprocess the data, MS/MS fragment ions within +/- 17 Da of the precursor m/z were removed. Subsequently, the MS/MS spectra were window filtered, selecting only the top 6 fragment ions in a +/- 50 Da window throughout the spectrum. The tolerance for MS/MS fragment ions was set to 0.5 Da, while the precursor ion mass tolerance was set to 2.0 Da. The molecular network was constructed based on specific criteria. Edges in the network were filtered to have a cosine score above 0.7 and a minimum of 6 matched peaks. Additionally, edges between two nodes were retained in the network if both nodes appeared in each other's respective top 10 most similar nodes. The maximum size of a molecular family was set to 100, and the lowest-scoring edges were removed from the molecular families until the family size fell below this threshold. The spectra within the network were then searched against the spectral libraries available in GNPS. The library spectra were filtered using the same criteria as the input data. Matches between the network spectra and library spectra were considered valid if they had a score above 0.7 and a minimum of 6 matched peaks. The resulting molecular network is available at https://gnps.ucsd.edu/ProteoSAFe/status.jsp?task=

[1232ebcc68ec4e918d1754ae396dc86a/](). The data output was imported into Cytoscape version 3.8.2 for visualization and analysis ([https://cytoscape.org](https://cytoscape.org)) [23].

## Statistical analysis

Survival and mortality data analysis was conducted using GraphPad Prism 8.3.4 software (GraphPad Software Inc., La Jolla, CA, USA). The median lethal time values ($LT_{50,90,99}$) and the median lethal concentration values ($LC_{50,90,99}$) were calculated by using LdP line statistical software, which is devoted to calculating probit and regression analysis according to [24]; additionally, it is used to illustrate the relation between stimulus and response in toxicological and biological studies. This software package was originally purchased from Prof Dr. Ehab Mostafa Bakr (Plant Protection Research Institute, Acarology Department, Cairo, Egypt). Dose-response log probit and regression analyses were performed according to [25]. Survival curves were calculated from the number of dead ticks recorded daily over the different concentrations from each plant extract using a Kaplan–Meier curve according to methods previously reported [26].

$$Survival\ (\%) = \frac{Number\ of\ survuved\ ticks}{Total\ number\ of\ ticks} \times 100 \tag{1}$$

The corrected mortality percent was calculated from the following equation, which was obtained from a previously reported study [27]:

$$Mortality\ rate\ (\%) = \frac{experimental\ group,\% -\ negative\ control\ group,\%}{100\% -\ negative\ control\ group,\%} \times 100 \tag{2}$$

They are marked in the figures with asterisks and defined in each corresponding figure legend, together with the name of the statistical test that was used.

## Results

### Acaricidal activity of plant extracts against *H. longicornis* ticks

Among the 11 plant extracts screened under high concentrations (50 mg/mL), both *A. judaica* and *F. tenacissima* exhibited potent effects against *H. longicornis* ticks, resulting in 100% mortality (5/5) of the ticks. *A. judaica* demonstrated potency within 48 h, while *F. tenacissima* showed dose-dependent mortality on a daily basis, reaching 100% mortality at 96 h posttreatment compared to untreated ticks. Cypermethrin, a positive control drug, induced 100% mortality of the ticks at 48 h posttreatment with a concentration of 5 mg/mL. On the other hand, the extract from *C. droserifolia* showed moderate to weak activity, with 60% mortality (3/5) observed in adult ticks at 96 h posttreatment. *O. baccatus* extract displayed a weak effect against the ticks, resulting in 20% mortality (1/5) (Table 2, Fig 1). In contrast, the other tested plant extracts, including *Trichodesma africanum*, *Anabasis setifera*, *Aerva javanica*, *Carthamus tinctorius*, *Citrullus colocynthis*, *Ocimum basilicum*, and *Origanum majorana*, did not exhibit any activity against the ticks. These extracts showed no mortality (0%, 0/5) even at a concentration of 50 mg/mL over 96 h of treatment (Trial 1, Table 2, Fig 1).

Among the 11 plant extracts tested under a moderate treatment concentration (25 mg/mL), only *A. judaica* exhibited a potent effect against *H. longicornis* ticks, resulting in 100% mortality (5/5) within 48 h (Trial 1, Table 2; Fig 1)

Extracts from *Cleome droserifolia*, *Origanum majorana*, and *A. setifera* plants show weak activity, with 20% mortality (1/5) observed in the ticks at 96 h posttreatment (Table 2; Fig 1).

**Table 2. The mortality rate of *Haemaphysalis longicornis* ticks after treatment with plant extracts (Trial 1).**

| Plant extract | Plant extract dose (mg/mL) | Number of surviving ticks/ day | | | | Survival % after 96 h treatment | Mortality rate (%) | | | |
|---|---|---|---|---|---|---|---|---|---|---|
| | | 24 h | 48 h | 72 h | 96 h | | 24 h | 48 h | 72 h | 96 h |
| *Artemisia judaica L.* | 50 | 5 | 0 | 0 | 0 | 0% | 0% | 100% | 100% | 100% |
| | 25 | 5 | 0 | 0 | 0 | 0% | 0% | 100% | 100% | 100% |
| | 10 | 5 | 1 | 1 | 0 | 0% | 0% | 80% | 80% | 100% |
| *Cleome droserifolia (Forssk.) Delile* | 50 | 5 | 4 | 3 | 2 | 40% | 0% | 20% | 40% | 60% |
| | 25 | 5 | 5 | 5 | 4 | 80% | 0% | 0% | 0% | 20% |
| | 10 | 5 | 5 | 5 | 5 | 100% | 0% | 0% | 0% | 0% |
| *Trichodesma africanum (L.) Sm.* | 50 | 5 | 5 | 5 | 5 | 100% | 0% | 0% | 0% | 0% |
| | 25 | 5 | 5 | 5 | 5 | 100% | 0% | 0% | 0% | 0% |
| | 10 | 5 | 5 | 5 | 5 | 100% | 0% | 0% | 0% | 0% |
| *Ochradenus baccatus Delile* | 50 | 5 | 5 | 4 | 4 | 80% | 0% | 0% | 20% | 20% |
| | 25 | 5 | 5 | 5 | 5 | 100% | 0% | 0% | 0% | 0% |
| | 10 | 5 | 5 | 5 | 5 | 100% | 0% | 0% | 0% | 0% |
| *Forsskaolea tenacissima L.* | 50 | 5 | 4 | 3 | 0 | 0% | 0% | 20% | 40% | 100% |
| | 25 | 5 | 5 | 5 | 5 | 100% | 0% | 0% | 0% | 0% |
| | 10 | 5 | 5 | 5 | 5 | 100% | 0% | 0% | 0% | 0% |
| *Anabasis setifera Moq.* | 50 | 5 | 5 | 5 | 5 | 100% | 0% | 0% | 0% | 0% |
| | 25 | 5 | 5 | 5 | 4 | 80% | 0% | 0% | 0% | 20% |
| | 10 | 5 | 5 | 5 | 5 | 100% | 0% | 0% | 0% | 0% |
| *Aerva javanica (Burm.f.) Juss. ex Schult.* | 50 | 5 | 5 | 5 | 5 | 100% | 0% | 0% | 0% | 0% |
| | 25 | 5 | 5 | 5 | 5 | 100% | 0% | 0% | 0% | 0% |
| | 10 | 5 | 5 | 5 | 5 | 100% | 0% | 0% | 0% | 0% |
| *Carthamus tinctorius L.* | 50 | 5 | 5 | 5 | 5 | 100% | 0% | 0% | 0% | 0% |
| | 25 | 5 | 5 | 5 | 5 | 100% | 0% | 0% | 0% | 0% |
| | 10 | 5 | 5 | 5 | 5 | 100% | 0% | 0% | 0% | 0% |
| *Citrullus colocynthis (L.) Schrad.* | 50 | 5 | 5 | 5 | 5 | 100% | 0% | 0% | 0% | 0% |
| | 25 | 5 | 5 | 5 | 5 | 100% | 0% | 0% | 0% | 0% |
| | 10 | 5 | 5 | 5 | 5 | 100% | 0% | 0% | 0% | 0% |
| *Ocimum basilicum L.* | 50 | 5 | 5 | 5 | 5 | 100% | 0% | 0% | 0% | 0% |
| | 25 | 5 | 5 | 5 | 5 | 100% | 0% | 0% | 0% | 0% |
| | 10 | 5 | 5 | 5 | 5 | 100% | 0% | 0% | 0% | 0% |
| *Origanum majorana L.* | 50 | 5 | 5 | 5 | 5 | 100% | 0% | 0% | 0% | 0% |
| | 25 | 5 | 4 | 4 | 4 | 80% | 0% | 20% | 20% | 20% |
| | 10 | 5 | 5 | 5 | 5 | 100% | 0% | 0% | 0% | 0% |
| Cypermethrin | 5 | 5 | 0 | 0 | 0 | 0% | 0% | 100% | 100% | 100% |
| | 1 | 5 | 0 | 0 | 0 | 0% | 0% | 100% | 100% | 100% |
| Negative control (80%-MeOH)[a] | 0 | 5 | 5 | 5 | 5 | 100% | 0% | 0% | 0% | 0% |
| | | 5 | 0 | 0 | 0 | 0% | 0% | 0% | 0% | 0% |
| | | 5 | 0 | 0 | 0 | 0% | 0% | 0% | 0% | 0% |

The mortality rate of *Haemaphysalis longicornis* ticks after treatment with plant extracts. Five ticks were used per concentration for each plant extract tested. The solvent used for each plant concentration as well as the positive and negative controls was Methanol 80% (80% MeOH)[a]. Tick survival and mortalities were observed until 96 hours posttreatment.

Under a low treatment concentration (10 mg/mL), *A. judaica* exhibited 80% mortality (4/5) within 48 h and 100% mortality (5/5) within 96 h posttreatment compared to the untreated ticks (Trial 1, Table 2, Fig 1).

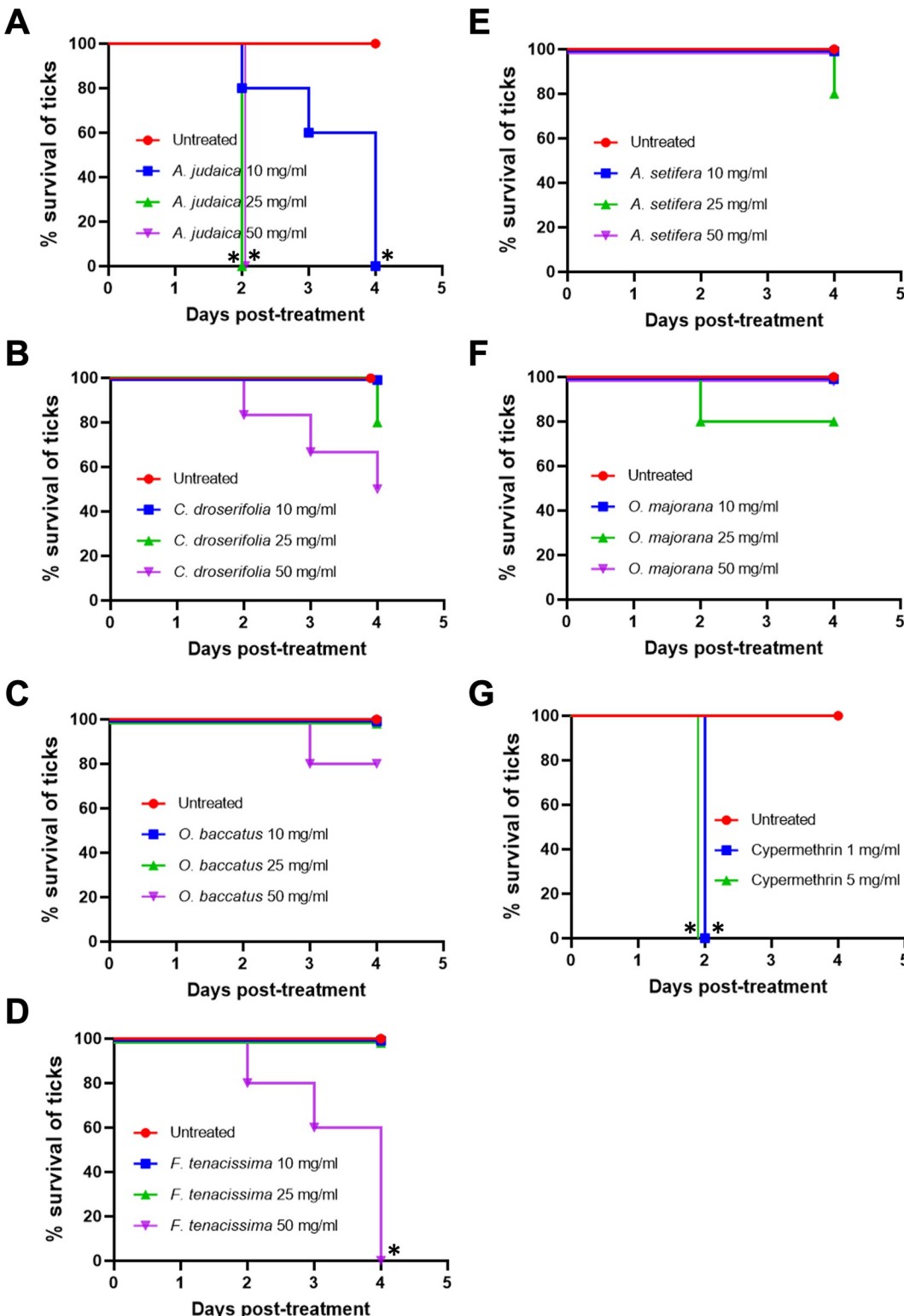

**Fig 1. Percent survival of ticks at 96 hours posttreatment with potent crude plant extracts.** The percent survival of ticks was monitored every 24 h over the different concentrations of each plant extract until 96 h posttreatment. Six plant extracts that showed potent, moderate, or low acaricidal effects in this study were as follows: *Artemisia judaica*, *Cleome droserifolia*, *Ochradenus baccatus*, *Forsskaolea tenacissima*, *Anabasis setifera*, and *Origanum majorana*. Cypermethrin was used as a reference drug, and the 80% methanol carrier solution was used as an untreated negative control. Three concentrations were

used per plant candidate (50, 25, and 10 mg/mL). Five ticks were used per plant extract concentration (N = 5); thus, the number of ticks used per plant extract was 15, while cypermethrin was used at concentrations of 1 and 5 mg/mL. The percent survival of ticks was monitored for each of the plant concentrations at 10, 25, and 50 mg/mL after 24, 48, 72, and 96 h posttreatment, as listed in Table 2. The survival of ticks treated with each plant extract concentration was monitored daily until 96 h posttreatment, as follows (survived/total number): (A) *A. judaica* (0/5,0%; 0/5,0%; 0/5,0%), (B) *C. droserifolia* (5/5,100%; 4/5,80%; 2/5,40%), (C) *O. baccatus* (5/5,100%; 5/5,100%; 4/5, 80%), (D) *F. tenacissima* (5/5,100%; 5/5,100%; 0/5, 0%), (E) *A. setifera* (5/5,100%; 4/5,80%; 5/5,100%), (F) *O. majorana* (5/5,100%; 4/5,80%; 5/5,100%), (G) Cypermethrin at 1 and 5 mg/mL (0/5,0%; 0/5, 0%), and untreated control group (5/5,0%; 5/5,0%; 5/5, 0%). Statistically significant differences in the survival were analyzed Log-rank (Mantel-cox) test (* *P*<0.05).

## Confirmation of the acaricidal activity of *A. judaica* plant extract against *H. longicornis* ticks

To confirm the activity of the most potent candidate, *A. judaica*, trial 2 was conducted using a larger number of ticks. *A. judaica* demonstrated potency within 48 h, in dose-dependent mortality on a daily basis, reaching 100% mortality (15/15), and 0% survival (0/15) at 96 h posttreatment compared to untreated ticks (Table 3, Fig 2). Under a moderate treatment concentration (25 mg/mL), *A. judaica* exhibited a potent effect against *H. longicornis* ticks with 100% mortality (15/15) within 72 h (Trial 2, Table 3; Fig 2) compared to the untreated ticks. Under the low dose, *A. judicia*'s efficacy in this concentration resulted in only 6.7% mortality (1/15) within 72 to 96 hours post-treatment compared to the untreated ticks (Trial 2, Table 3 and Fig 2), indicating that this extract exhibited a weak effect at the low dose.

## Calculation of median lethal concentration 50 ($IC_{50}$) and median lethal time 50 ($LT_{50}$) for the potent candidate plant extracts against *H. longicornis* ticks

The median lethal times ($LT_{50}$ and $LT_{90}$) of the potent anti-tick candidates, *A. judaica*, *C. droserifolia*, and *F. tenacissima* plant extracts, were calculated. For *A. judaica* at a concentration of 10 mg/mL, the $LT_{50}$ and $LT_{90}$ values were determined to be 43.19 and 70.38 hours, respectively (Trial 1). However, at higher concentrations, the median lethal time could not be determined due to the strong lethal effect of *A. judaica* plant extracts and the low number of ticks used (Trial 1) (Table 4). In trial 2, by using a larger number of ticks, the $LT_{50}$ and $LT_{90}$ values of *A. judaica* were determined to be 30.37, 55.08, and 37.11, 61.93 hours at the high and moderate doses, respectively (Table 5).

For the *C. droserifolia* plant extract, the $LT_{50}$ and $LT_{90}$ values were 82.21 and 173.06 hours, respectively, at a concentration of 50 mg/mL. At a lower concentration of 25 mg/mL, the $LT_{50}$ and $LT_{90}$ values were 190.0 and 435.09 hours, respectively (Table 4). In the case of the *F. tenacissima* plant extract, which shows efficacy at a higher dose of 50 mg/mL, the $LT_{50}$ and $LT_{90}$ values were calculated as 66.27 and 100.57 hours, respectively (Table 4).

**Table 3. The mortality rate of *Haemaphysalis longicornis* ticks after treatment with *A. judaica* extract (Trial 2).**

| Plant extract | Plant extract dose (mg/mL) | Number of surviving ticks/day | | | | Survival % after 96 h treatment | Mortality rate (%) | | | |
|---|---|---|---|---|---|---|---|---|---|---|
| | | 24 h | 48 h | 72 h | 96 h | | 24 h | 48 h | 72 h | 96 h |
| *Artemisia judaica L.* | 50 | 14 | 0 | 0 | 0 | 0% | 6.7% | 100% | 100% | 100% |
| | 25 | 15 | 14 | 0 | 0 | 0% | 0% | 6.7% | 100% | 100% |
| | 10 | 15 | 15 | 14 | 14 | 93.3% | 0% | 0% | 6.7% | 6.7% |
| **Cypermethrin** | 5 | 0 | 0 | 0 | 0 | 0% | 100% | 100% | 100% | 100% |
| | 1 | 0 | 0 | 0 | 0 | 0% | 100% | 100% | 100% | 100% |
| Negative control (80%-MeOH)[a] | 0 | 15 | 15 | 15 | 15 | 100% | 0% | 0% | 0% | 0% |

The mortality rate of *Haemaphysalis longicornis* ticks after treatment with plant extracts. Fifteen ticks were used per concentration from *Artemisia judaica* plant extract. The solvent used for each plant concentration as well as the positive and negative controls was Methanol 80% (80% MeOH)[a]. Tick survival and mortalities were observed until 96 hours posttreatment.

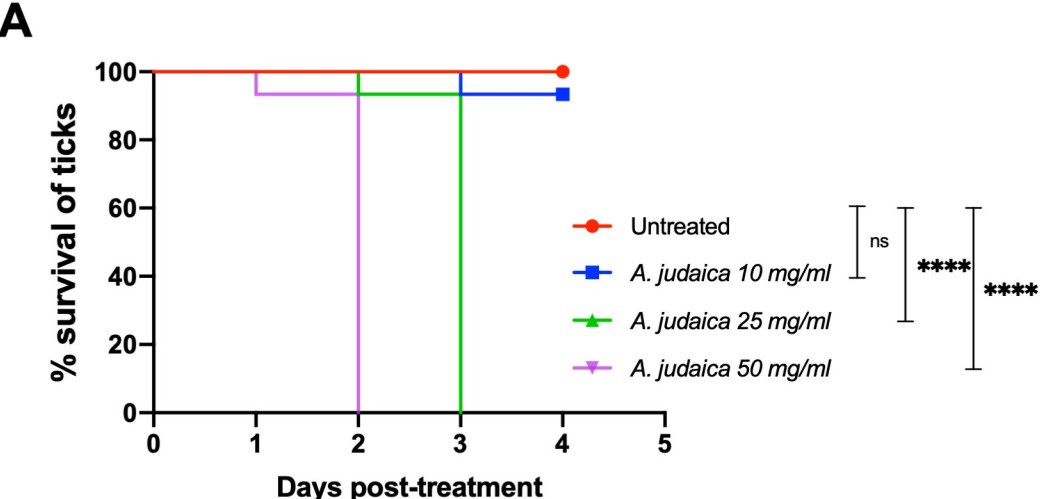

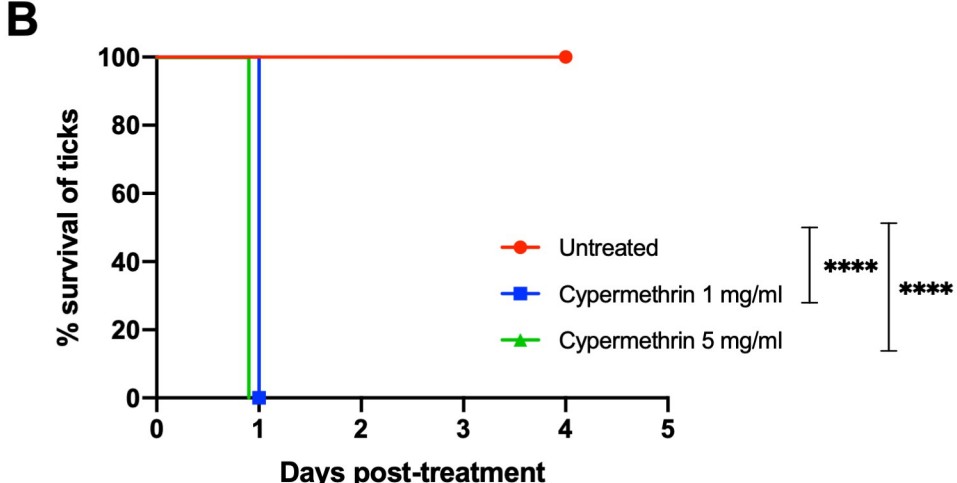

**Fig 2. Percent survival of ticks at 96 hours posttreatment with *Artemisia judaica* crude plant extracts.** The percent survival of ticks was monitored every 24 h over the different concentrations of *A. judaica* plant extract until 96 h posttreatment. Cypermethrin was used as a reference drug, and the 80% methanol carrier solution was used as an untreated negative control. Three concentrations were used per plant candidate (50, 25, and 10 mg/mL). Fifteen ticks were used per plant extract concentration (N = 15); thus, the number of ticks used per plant extract was 45, while cypermethrin was used at concentrations of 1 and 5 mg/mL. The percent survival of ticks was monitored for each of the plant concentrations at 10, 25, and 50 mg/ml after 24, 48, 72, and 96 h posttreatment, as listed in Table 3. The survival of ticks treated with each plant extract concentration was monitored daily until 96 h posttreatment, as follows (survived/total number): **(A)** *A. judaica* (14/15, 0%; 0/15, 0%; 0/15, 0%), **(B)** Cypermethrin at 1 and 5 mg/mL (0/15, 0%; 0/15, 0%), and untreated control group (15/15, 0%; 15/15, 0%; 15/15, 0%). Statistically significant differences in the survival were analyzed by Log-rank (Mantel-cox) test (* $P<0.05$).

The median lethal concentration ($LC_{50}$) of the potent anti-tick candidates, *A. judaica*, *C. droserifolia*, and *F. tenacissima* plant extracts, were calculated. *Artemisia judaica* exhibits the highest efficacy among all tested plant extracts through the lower $LC_{50}$ values. The $LC_{50}$ values of both *C. droserifolia* and *F. tenacissima* plant extracts were 42.89 mg/mL and 34.05 mg/mL, respectively (Trial 1 and Table 4), which is higher than that determined from *A. judaica* 13.73 mg/mL (Trial 2 and Table 5).

**Table 4. Median lethal time and concentration of the Egyptian plant extracts against adult female *H. longicornis* ticks (Trial 1).**

| Plant extract | Concentration (mg/mL) | Median lethal time 50 $(LT_{50})$[a]/hour | Lethal time 90 $(LT_{90})$[b]/hour | r[c] | Chi-square $(\chi 2)$[d] | Slope ± | Median lethal concentration 50 $(LC_{50})$[e] | Concentration 90 $(LC_{90})$[f] | Concentration 99 $(LC_{99})$[g] | r | Chi-square $(\chi 2)$ | Slope ± | P value |
|---|---|---|---|---|---|---|---|---|---|---|---|---|---|
| *Artemisia judaica* L. | 50 | ND | ND | ---- | ---- | ---- | ND | 0.0074 | 2.55 | 0.9641 | 0.0001 | 0.4102 ±1.24 | 0.990 |
| | 25 | ND | ND | ---- | ---- | ---- | | | | | | | |
| | 10 | 43.19 | 70.38 | 0.9603 | 35.25 | 6.04 ±0.48 | | | | | | | |
| *Cleome droserifolia* (Forssk.) Delile | 50 | 82.21 | 173.06 | 0.9965 | 0.7109 | 3.96± 0.46 | 42.89 | 92.30 | 172.39 | 0.9992 | 0.1378 | 3.8504 ±0.48 | 0.710 |
| | 25 | 190.0 | 435.09 | 0.6563 | 21.03 | 3.56 ±0.97 | | | | | | | |
| | 10 | ND | ND | ---- | ---- | ---- | | | | | | | |
| *Forsskaolea tenacissima* L. | 50 | 66.27 | 100.57 | 0.9239 | 39.79 | 7.07 ±0.68 | 34.05 | 40.39 | 46.42 | 0.8448 | 0.0 | 17.285 ±3.10 | 1 |
| | 25 | ND | ND | ---- | ---- | ---- | | | | | | | |
| | 10 | ND | ND | ---- | ---- | ---- | | | | | | | |
| Positive control (cypermethrin) | 5 | ND | ND | ---- | ---- | ---- | ND | ND | ND | ---- | ---- | ---- | ---- |
| | 1 | ND | ND | ---- | ---- | ---- | ND | ND | ND | ---- | ---- | ---- | ---- |
| **Negative control (80% MeOH)** | 0 | ND | ND | ---- | ---- | ---- | ND | ND | ND | ---- | ---- | ---- | ---- |

The median lethal times (50 and 90) and the median lethal concentrations (50, 90, and 99) were calculated with the LdP line statistical software after 96 h posttreatment with each potent plant extract. Data were analyzed by the Ldp line program using dose–response log probit and regression analyses. ND: not determined. [a]$TL_{50}$: Median lethal time 50; [b]$TL_{90}$: Median lethal time 90;[c] r: Regression coefficient;[d] $\chi 2$: Chi-square; [e]$LC_{50}$: Median lethal concentration 50; [f] $LC_{90}$: Median lethal concentration 90; [g] $LC_{99}$: Median lethal concentration 99; 80% MeOH: 80% methanolic. *P-value* was calculated from the analysis conducted with Ldp line software.

## Metabolite profiling of the *A. judaica* leaf extract based on LC–MS

The metabolite profiling of the crude extract from *A. judaica* was carried out via LC–MS/MS. The total ion chromatogram of the extract is shown in Fig 3. $AutoMS^2$ was then performed, choosing the most prevalent $MS^1$ ions for $MS^2$ fragmentation. Finally, the chemical

**Table 5. Median lethal time and concentration of *Artemisia judaica* plant extracts against adult female *H. longicornis* ticks (Trial 2).**

| Plant extract | Concentration (mg/mL) | Median lethal time 50 $(LT_{50})$[a]/hour | Lethal time 90 $(LT_{90})$[b]/hour | r[c] | Chi-square $(\chi 2)$[d] | Slope ± | Median lethal concentration 50 $(LC_{50})$[e] | Concentration 90 $(LC_{90})$[f] | Concentration 99 $(LC_{99})$[g] | r | Chi-square $(\chi 2)$ | Slope (±) | P value |
|---|---|---|---|---|---|---|---|---|---|---|---|---|---|
| *Artemisia judaica* L. | 50 | 30.37 | 37.11 | 0.8811 | 0.00 | 14.72 ±2.97 | 13.73 | 17.97 | 22.38 | 0.9032 | 0.00 | 10.97 (2.08) | 1 |
| | 25 | 55.08 | 61.93 | 0.9232 | 0.00 | 25.17 ±5.08 | | | | | | | |
| | 10 | 278.50 | 746.95 | 0.8604 | 3.2474 | 2.99 ±1.08 | | | | | | | |

The median lethal times (50 and 90) and the median lethal concentrations (50, 90, and 99) were calculated with the LdP line statistical software after 96 h posttreatment with each potent plant extract. Data were analyzed by the Ldp line program using dose-response log probit and regression analyses. ND: not determined. [a]$TL_{50}$: Median lethal time 50; [b]$TL_{90}$: Median lethal time 90;[c] r: Regression coefficient; $\chi 2$: Chi-square; [e] $LC_{50}$: Median lethal concentration 50; [f] $LC_{90}$: Median lethal concentration 90; [g] $LC_{99}$: Median lethal concentration 99; 80% MeOH: 80% methanolic. *P-value* was calculated from the analysis conducted with Ldp line software.

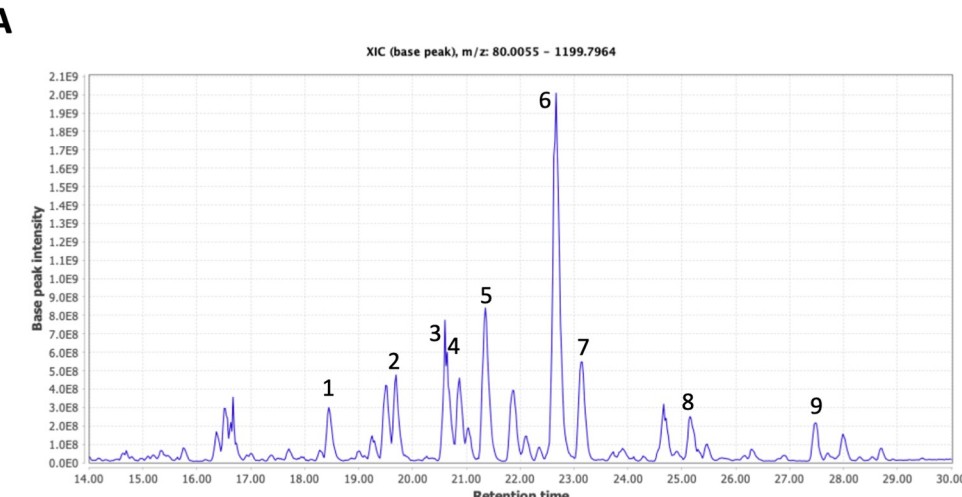

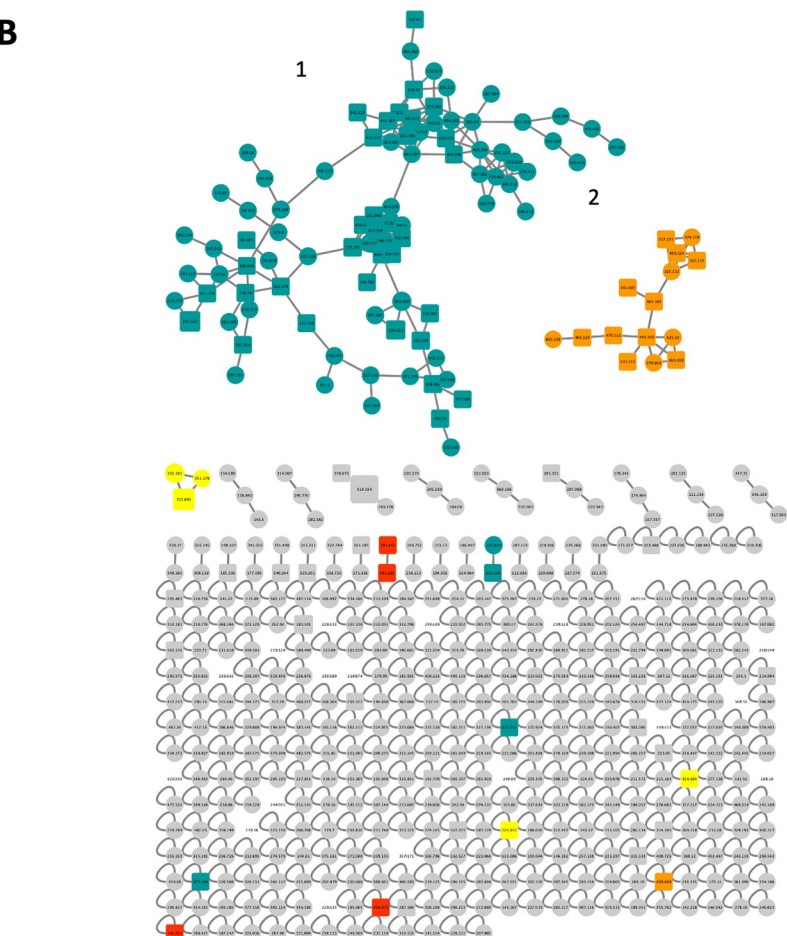

**Fig 3. Total ion chromatogram (TIC) of the *A. judaica* extract. (A)** The number above each peak represents the peak numbers corresponding to the peak numbers in Table 6. **(B)** Annotation of the molecular networking of the norlignanes and phenolic glycosides derived from *A. judaica* leaf extracts obtained from the GNPS spectral library. The green-colored structures (terpenoids and steroids), orange-colored structures (phenylpropanoids and flavonoids-glycosides), yellow-colored structures (flavonoids), red-colored structures (benzenoids), and gray-colored structures (nonmatched) from the leaf crude extract of the *A. judaica* plant.

**Table 6. Tentative identification of compounds from the crude extract of *A. judaica* by LC–MS/MS in the positive ion mode.**

| Peak Number | RT (min) | *m/z* Detected | Exact Mass | Molecular Formula | Adduct | Ion Tentative Identification | Compound Class |
|---|---|---|---|---|---|---|---|
| 1 | 18.45 | 167.107 | 184.110 | $C_{10}H_{16}O_3$ | $[M-H_2O+H]^+$ | Gallicynoic acid I | Hydroxy fatty acid |
| 2 | 19.69 | 139.112 | 138.105 | $C_9H_{14}O$ | $[M+H]^+$ | Pulegenone | Cyclic ketone |
| 3 | 20.63 | 167.107 | 166.100 | $C_{10}H_{14}O_2$ | $[M+H]^+$ | 6-Oxochamphor | Terpenoid |
| 4 | 20.87 | 235.169 | 252.173 | $C_{15}H_{24}O_3$ | $[M-H_2O+H]^+$ | Lippidulcine A | Terpenoid |
| 5 | 21.35 | 315.086 | 314.079 | $C_{17}H_{14}O_6$ | $[M+H]^+$ | 5,7-Dihydroxy-8,4'-dimethoxyisoflavone | Flavonoid |
| 6 | 22.66 | 153.128 | 152.120 | $C_{10}H_{16}O$ | $[M+H]^+$ | Camphor | Terpenoid |
| 7 | 23.14 | 315.086 | 314.079 | $C_{17}H_{14}O_6$ | $[M+H]^+$ | 4',5-Dihydroxy-3',7-dimethoxyisoflavone | Flavonoid |
| 8 | 25.16 | 125.096 | 124.089 | $C_8H_{12}O$ | $[M+H]^+$ | 6-Methyl-3,5-heptadien-2-one | Oxygenated hydrocarbon |
| 9 | 27.46 | 237.185 | 236.178 | $C_{15}H_{24}O_2$ | $[M+H]^+$ | Artemone | Terpenoid |

components of the crude extracts of *A. judaica* were putatively identified through manual examination of the resulting MS/MS spectra and are reported in Table 6 as putative identifications.

The compounds were annotated using databases including PubChem (https://pubchem.ncbi.nlm.nih.gov/compound), the UC2 database (including KNApSAcK (http://kanaya.naist.jp/KNApSAcK/), the Human Metabolome Database (http://www.hmdb.ca), and the Dictionary Natural Product (https://dnp.chemnetbase.com/).

A total of 9 substances were detected and putatively identified, as shown in Table 6. These metabolites are members of the following eight families of natural compounds: hydroxy fatty acids, cyclic ketones, terpenoids, flavonoids, and oxygenated hydrocarbons.

## Molecular network of the *A. judaica* leaf extract based on LC–MS

In addition to manual annotation, we analyzed the crude extract of *A. judaica* using the molecular networking (MN) approach using the GNPS website (http://gnps.ucsd.edu). The MN facilitates data analysis by clustering the LC–MS/MS spectra based on fragmentation cosine similarities [28]. The MN of the extract was generated from the LC–MS/MS analysis data to analyze the metabolic content of *A. judaica* (Figs 3B and 4). The MN generated from the leaf extract of *A. judaica* showed several clusters; each cluster shared some distinct fragments and fragmentation patterns. The detected compounds analyzed by LC–MS/MS for *A. judaica* L. are shown in Table 6. The network identified two major clusters comprising terpenoids and steroids, phenylpropanoids, flavonoid-glycosides, and other clusters of flavonoids, benzenoids, and other non-matched compounds (Fig 4 and Table 7). The mass spectrometry-based MN allowed for the identification and putative annotation of 74 compounds (Table 7). Camphor showed the highest base peak intensity in the LC–MS analysis (Fig 3A, peak number 6; Tables 6 and 7).

## Discussion

The screening of 11 plant extracts revealed that a crude methanolic extract from the *A. judaica* plant exhibited potential activity against adult female ticks of *H. longicornis*. In addition, a previous study illustrated the acaricidal effects of a botanical extract from *Eupatorium adenophorum* against various growth stages of *H. longicornis*. However, the concentrations used in that study (0.5, 1, and 1.5 g/mL) were considerably higher than those used in our investigation. Specifically, a concentration of 1.5 g/ml achieved a 100% acaricidal effect on both larval and nymphal stages, whereas a concentration of 1 g/mL resulted in 100% larval mortality and 93% mortality in nymphs within 6 hours posttreatment [29]. Furthermore, oleoresin derived from

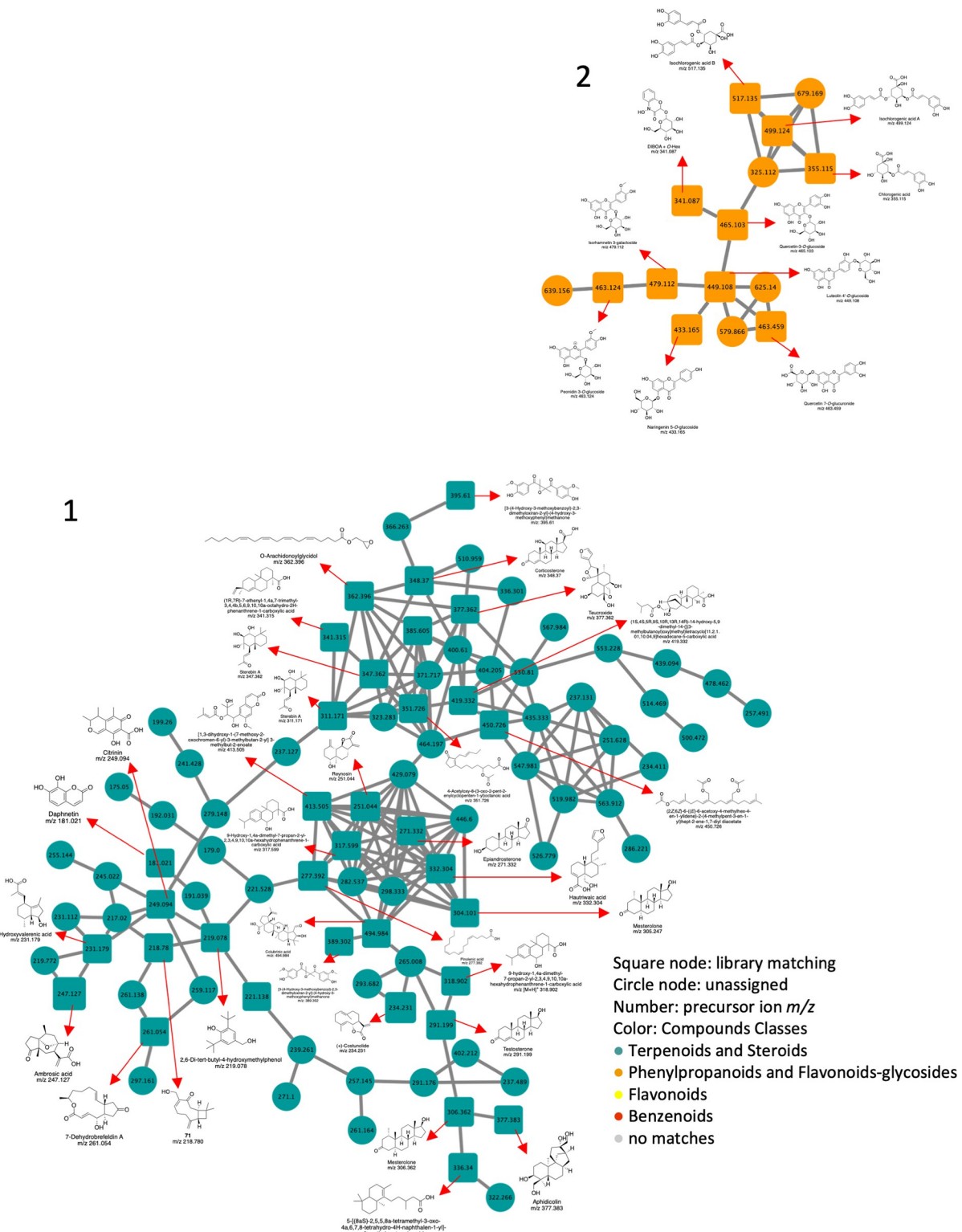

**Fig 4. Magnification and annotation of the molecular network, Clusters 1 and 2, of the leaf crude extract of *A. judaica*.**

**Table 7. Detected compounds analyzed by LC–MS/MS for the *Artemisia judaica* L. crude extract.**

| No. | Compound identification | Compound class | RT[a] | Formula | Identification confidence | m/z Observed |
|---|---|---|---|---|---|---|
| 1 | 1-[3-Hydroxy-2-(2-hydroxy-2-propanyl)-2,3-dihydro-1-benzofuran-5-yl] ethanone | Acetophenones | 14.41 | $C_{13}H_{16}O_4$ | L1 | 236.975 |
| 2 | Lycopsamine N-oxide | Alkaloids | 10.24 | $C_{15}H_{25}NO_6$ | L2 | 315.133 |
| 3 | Kainic acid | Amino acids, peptides, and analogs | 10.01 | $C_{10}H_{15}NO_4$ | L2 | 214.904 |
| 4 | Kynurenine | Amino acids, peptides, and analogs | 9.09 | $C_{10}H_{12}N_2O_3$ | L2 | 209.849 |
| 5 | L-Tryptophan | Amino acids, peptides, and analogs | 10.96 | $C_{11}H_{12}N_2O_2$ | L2 | 205.463 |
| 6 | L-Tyrosine | Amino acids, peptides, and analogs | 4.66 | $C_9H_{11}NO_3$ | L2 | 181.709 |
| 7 | Na,Na-Dimethylhistidine | Amino acids, peptides, and analogs | 0.06 | $C_8H_{13}N_3O_2$ | L2 | 183.105 |
| 8 | Val-Leu | Amino acids, peptides, and analogs | 10.47 | $C_{11}H_{22}N_2O_3$ | L2 | 229.609 |
| 9 | Bis(2-ethylhexyl) benzene-1,2-dicarboxylate | Benzoic acids and derivatives | 0.02 | $C_{24}H_{38}O_4$ | L2 | 391.283 |
| 10 | DIBOA + O-Hex | Carbohydrates and carbohydrate conjugates | 10.07 | $C_{14}H_{17}NO_9$ | L2 | 341.087 |
| 11 | [1,3-dihydroxy-1-(7-methoxy-2-oxochromen-6-yl)-3-methylbutan-2-yl] 3-methylbut-2-enoate | Coumarins | 16.04 | $C_{20}H_{24}O_7$ | L2 | 413.505 |
| 12 | 7-[(2S,3R,4S,5S,6R)-3,4,5-trihydroxy-6-(hydroxymethyl)oxan-2-yl] oxychromen-2-one | Coumarins | 11.36 | $C_{15}H_{16}O_8$ | L1 | 324.884 |
| 13 | Daphnetin | Coumarins | 12.71 | $C_9H_6O_4$ | L2 | 181.021 |
| 14 | [(2E,6Z,10Z)-7,11-bis(acetyloxymethyl)-3,15-dimethylhexadeca-2,6,10,14-tetraenyl] acetate | Fatty acids and conjugates | 20.24 | $C_{26}H_{40}O_6$ | L2 | 450.726 |
| 15 | 4-Acetyloxy-8-(3-oxo-2-pent-2-enylcyclopenten-1-yl)octanoic acid | Fatty acids and conjugates | 17.03 | $C_{20}H_{30}O_5$ | L2 | 351.726 |
| 16 | Azelaic acid | Fatty acids and conjugates | 13.07 | $C_9H_{16}O_4$ | L2 | 227.852 |
| 17 | Icos-19-ene-1,2,4-triol | Fatty acids and conjugates | 28.18 | $C_{20}H_{40}O_3$ | L2 | 329.201 |
| 18 | *O*-Arachidonoylglycidol | Fatty acids and conjugates | 11.45 | $C_{23}H_{36}O_3$ | L2 | 362.396 |
| 19 | Pinolenic acid | Fatty acids and conjugates | 12.98 | $C_{18}H_{30}O_2$ | L2 | 277.392 |
| 20 | 5-Hydroxy-2',4',7,8-Tetramethoxyflavone | Flavonoids | 20.62 | $C_{19}H_{18}O_7$ | L2 | 359.073 |
| 21 | 5,7-Dihydroxy-2-(3-hydroxy-4-methoxyphenyl)-2,3-dihydrochromen-4-one | Flavonoids | 15.41 | $C_{16}H_{14}O_6$ | L2 | 303.086 |
| 22 | Cirsimaritin | Flavonoids | 21.21 | $C_{17}H_{14}O_6$ | L2 | 314.934 |
| 23 | Isorhamnetin 3-galactoside | Flavonoids | 14.13 | $C_{22}H_{22}O_{12}$ | L2 | 479.112 |
| 24 | Luteolin | Flavonoids | 14.30 | $C15H10O_6$ | L2 | 287.106 |
| 25 | Luteolin 4'-*O*-glucoside | Flavonoids | 13.58 | $C_{21}H_{20}O_{11}$ | L2 | 449.108 |
| 26 | Luteonin 7-glucuronide | Flavonoids | 14.27 | $C_{21}H_{18}O_{12}$ | L1 | 463.459 |
| 27 | Naringenin 5-*O*-glucoside | Flavonoids | 14.45 | $C_{21}H_{22}O_{10}$ | L1 | 433.165 |
| 28 | Nepetin | Flavonoids | 23.06 | $C_{16}H_{12}O_7$ | L1 | 315.091 |
| 29 | Peonidin 3-*O*-glucoside | Flavonoids | 14.70 | $C_{22}H_{23}O_{11}^+$ | L2 | 463.124 |
| 30 | Quercetin-3-*O*-glucoside | Flavonoids | 13.43 | $C_{21}H_{20}O_{12}$ | L2 | 465.103 |
| 31 | Vicenin-3 | Flavonoids | 12.55 | $C_{26}H_{28}O_{14}$ | L2 | 565.156 |
| 32 | Chlorogenic acid | Hydroxycinnamic acids and derivatives | 10.83 | $C_{16}H_{18}O_9$ | L1 | 355.115 |
| 33 | Isochlorogenic A | Hydroxycinnamic acids and derivatives | 14.60 | $C_{25}H_{24}O_{12}$ | L1 | 499.124 |
| 34 | Phellopterin (isochlorogenic acid B) | Hydroxycinnamic acids and derivatives | 10.49 | $C_{25}H_{24}O_{12}$ | L2 | 517.135 |

*(Continued)*

**Table 7.** (Continued)

| No. | Compound identification | Compound class | RT[a] | Formula | Identification confidence | m/z Observed |
|---|---|---|---|---|---|---|
| 35 | Sinapic acid | Hydroxycinnamic acids and derivatives | 11.22 | $C_{11}H_{12}O_5$ | L2 | 226.915 |
| 36 | Hydrocotarnine | Isoquinolines | 10.67 | $C_{12}H_{15}NO_3$ | L2 | 223.050 |
| 37 | 7-Dehydrobrefeldin A | Lactone Macrolide | 14.90 | $C_{16}H_{22}O_4$ | L1 | 261.054 |
| 38 | [3-(4-Hydroxy-3-methoxybenzoyl)-2,3-dimethyloxiran-2-yl]-(4-hydroxy-3-methoxyphenyl)methanone | Lignins | 14.18 | $C_{20}H_{20}O_7$ | L2 | 389.302 |
| 39 | 2-Naphthalenol | Naphthols derivatives | 18.32 | $C_{10}H_8O$ | L2 | 146.932 |
| 40 | [3-(4-Hydroxy-3-methoxybenzoyl)-2,3-dimethyloxiran-2-yl]-(4-hydroxy-3-methoxyphenyl)methanone | Phenolics | 18.11 | $C_{20}H_{20}O_7$ | L2 | 395.610 |
| 41 | 2-(Hydroxymethyl)-6-[5-[3-(hydroxymethyl)-5-(3-hydroxypropyl)-7-methoxy-2,3-dihydro-1-benzofuran-2-yl]-2-methoxyphenoxy]oxane-3,4,5-triol | Phenolics | 14.04 | $C_{26}H_{34}O_{11}$ | L2 | 540.244 |
| 42 | 2,6-Di-tert-butyl-4-hydroxymethylphenol | Phenolics | 11.85 | $C_{15}H_{24}O_2$ | L2 | 219.078 |
| 43 | 2,6-Di-tert-butyl-4-hydroxymethylphenol | Phenolics | 14.92 | $C_{15}H_{24}O_2$ | L2 | 221.138 |
| 44 | Dioctyl phthalate | Phenolics | 0.06 | $C_{24}H_{38}O_4$ | L2 | 391.452 |
| 45 | Feruloyltyramine | Phenolics | 17.32 | $C_{18}H_{19}NO_4$ | L2 | 314.393 |
| 46 | Citrinin | Polyketide | 13.00 | $C_{13}H_{14}O_5$ | L2 | 249.094 |
| 47 | Coprostanone | Steroids | 14.64 | $C_{27}H_{46}O$ | L2 | 385.605 |
| 48 | Corticosterone | Steroids | 14.22 | $C_{21}H_{30}O_4$ | L2 | 348.370 |
| 49 | Epiandrosterone | Steroids | 19.17 | $C_{19}H_{30}O_2$ | L2 | 275.164 |
| 50 | Mesterolone | Steroids | 16.06 | $C_{20}H_{32}O_2$ | L1 | 304.101 |
| 51 | Mesterolone | Steroids | 12.19 | $C_{20}H_{32}O_2$ | L1 | 306.362 |
| 52 | Testosterone | Steroids | 14.05 | $C_{19}H_{28}O_2$ | L1 | 291.199 |
| 53 | (-)-Isolongifolol | Terpenoids | 28.58 | $C_{15}H_{26}O$ | L2 | 205.121 |
| 54 | (1R,4Z,9S)-4-(Hydroxymethyl)-11,11-dimethyl-8-methylenebicyclo[7.2.0]undec-4-en-3-one | Terpenoids | 10.33 | $C_{15}H_{22}O_2$ | L1 | 218.780 |
| 55 | (1R,7R)-7-ethenyl-1,4a,7-trimethyl-3,4,4b,5,6,9,10,10a-octahydro-2H-phenanthrene-1-carboxylic acid | Terpenoids | 12.42 | $C_{20}H_{30}O_2$ | L2 | 341.315 |
| 56 | (1S,4S,5R,9S,10R,13R,14R)-14-hydroxy-5,9-dimethyl-14-{[(3-methylbutanoyl)oxy]methyl}tetracyclo[11.2.1.0$^{1,10}$.0$^{4,9}$]hexadecane-5-carboxylic acid | Terpenoids | 19.67 | $C_{25}H_{40}O_5$ | L2 | 419.332 |
| 57 | (9a-Hydroxy-3,8a-dimethyl-5-methylidene-2-oxo-4,4a,6,7,8,9-hexahydrobenzo[f][1]benzofuran-8-yl) acetate | Terpenoids | 13.51 | $C_{17}H_{22}O_5$ | L2 | 327.725 |
| 58 | 2-Naphthaleneacetic acid, 1,2,3,4,4a,5,8,8a-octahydro-1,8-dihydroxy-alpha,4a,8-trimethyl-5-oxo-, methyl ester | Terpenoids | 17.84 | $C_{16}H_{24}O_5$ | L1 | 279.141 |
| 59 | 3a-Hydroxy-3,5a,9-trimethyl-3,4,5,6,7,9b-hexahydrobenzo[g][1]benzofuran-2,8-dione | Terpenoids | 3.29 | $C_{15}H_{20}O_4$ | L2 | 281.051 |
| 60 | 5-[(8aS)-2,5,5,8a-tetramethyl-3-oxo-4a,6,7,8-tetrahydro-4H-naphthalen-1-yl]-3-methylpentanoic acid | Terpenoids | 15.19 | $C_{20}H_{32}O_3$ | L2 | 336.340 |
| 61 | 9-Hydroxy-1,4a-dimethyl-7-propan-2-yl-2,3,4,9,10,10a-hexahydrophenanthrene-1-carboxylic acid | Terpenoids | 18.75 | $C_{20}H_{28}O_3$ | L2 | 318.902 |
| 62 | Alpha-Bisabolol | Terpenoids | 19.20 | $C_{15}H_{26}O$ | L2 | 203.488 |
| 63 | Ambrosic acid | Terpenoids | 3.11 | $C_{15}H_{20}O_4$ | L1 | 247.127 |
| 64 | Aphidicolin | Terpenoids | 19.06 | $C_{20}H_{34}O_4$ | L2 | 377.383 |
| 65 | Camphora | Terpenoids | 1.50 | $C_{10}H_{16}O$ | L2 | 153.226 |
| 66 | Colubrinic acid | Terpenoids | 17.11 | $C_{30}H_{46}O_4$ | L2 | 494.984 |
| 67 | Costunolide | Terpenoids | 10.98 | $C_{15}H_{20}O_2$ | L2 | 234.231 |
| 68 | Hautriwaic acid | Terpenoids | 13.07 | $C_{20}H_{28}O_4$ | L1 | 332.304 |
| 69 | Hydroxyvalerenic acid | Terpenoids | 12.64 | $C_{15}H_{22}O_3$ | L1 | 231.179 |
| 70 | Hydroxyvalerenic acid | Terpenoids | 17.20 | $C_{15}H_{22}O_3$ | L1 | 235.136 |

(Continued)

**Table 7.** (Continued)

| No. | Compound identification | Compound class | RT$^a$ | Formula | Identification confidence | m/z Observed |
|-----|-----|-----|-----|-----|-----|-----|
| 71 | Reynosin | Terpenoids | 11.73 | $C_{15}H_{20}O_3$ | L2 | 251.044 |
| 72 | Sterebin A | Terpenoids | 14.37 | $C_{18}H_{30}O_4$ | L2 | 311.171 |
| 73 | Sterebin A | Terpenoids | 13.41 | $C_{18}H_{30}O_4$ | L2 | 347.362 |
| 74 | Teucroxide | Terpenoids | 14.91 | $C_{20}H_{26}O_7$ | L2 | 377.362 |

Annotated compounds based on LC–MS/MS and MN in the extract from *A. judaica* plants. The table shows the peak number, proposed compound, compound class and formula, retention time in minutes (RT)$^a$, and identification confidence. Level of confidence as proposed by [28]; L1: Structure confirmed by the reference standard or structure elucidation by NMR spectroscopy; L2: probable structure by the library spectrum match.

the *copaiba* tree, *Copaifera reticulata*, exhibits an antilarval effect against cattle tick (*Rhipicephalus (Boophilus) microplus*) with $LC_{50}$ and $LC_{99}$ values of 1.58 mg/mL and 3.49 mg/mL, respectively [27]. In contrast, our study estimated the $LC_{90}$ and $LC_{99}$ values of *A. judaica* to be 0.0074 and 2.55 mg/mL (Trial 1 and Table 4), while in trial 2, the $LC_{50}$, $LC_{90}$, $LC_{99}$ values of *A. judaica* were determined to be 13.73, 17.97, and 22.38 mg/mL respectively, against adult female ticks (Trial 2 and Table 5).

Plant-derived compounds with acaricidal activities have been evaluated from various plant families, including *Caesalpiniaceae, Ericaceae, Winteraceae, Solanaceae, Phytolaccaceae, Apiaceae, Myrtaceae, Meliaceae, Rutaceae, Amaryllidaceae, Euphorbiaceae,* and *Bromeliaceae* [30–36]. These plant extracts and metabolites have shown acaricidal effects against different species of ticks, such as *Rhipicephalus (Boophilus) microplus* [30, 31], *Rhipicephalus turanicus* (Acari: Ixodidae) [32], pyrethroid-resistant *Rhipicephalus (Boophilus) microplus* [34], and *Amblyomma variegatum* (Fabricius) (Acari: Ixodidae) [35].

In previous studies, the acaricidal effect of compounds or plant extracts was evaluated using the larval packet test (LPT) and larval immersion test (LIT), which mainly focused on the different tick species (FAO, 2004). Recently, modifications have been made to these tests to improve their repeatability [37]. In our study, we utilized an assay that allowed for the exposure of the ticks to the compound or extracts for an extended period of time (96 h) posttreatment under a consistent incubation temperature of 25±1˚C.

*A. judaica* plant extract collected from the Saudi Arabia-Jordan border region has recently been reported to exhibit antifungal activity against yeasts, *dermatophytes*, and *Aspergillus* strains [38]. Chemical analysis revealed that the main components of the extract were monoterpenes, accounting for 68.7% of the composition. Among the monoterpenes, piperitone was found to be the most abundant at 30.4%, followed by camphor at 16.1% and ethyl cinnamate at 11.0% [38]. In our study, *A. judaica* collected from the Egyptian Desert was found by LC–MS analysis to have camphor as the most abundant chemical constituent (10.38%) among the identified compounds. The differences in the chemical constituents of the same plant may be explained by its collection from different areas and being subjected to different chromatographic analyses.

The biological activities have also been reported of essential oils extracted from *A. judaica*, such as anthelmintic, anti-inflammatory, analgesic, and antipyretic activities [39]. Recently, the crude methanolic leaf extract from *A. judaica* has shown strong activity against *Toxoplasma gondii* and *Neospora caninum* parasite growth *in vitro* [13] and against human malaria *Plasmodium falciparum* 3D7 *in vitro* [20]. In the current study, *A. judaica* showed strong time- and concentration-dependent activities against hard adult female ticks of *H. longicornis*, suggesting the urgent need to test the efficacy of this plant extract in the treatment of tick infestation in vertebrates' models and tick environmental eradication.

*A. judaica* has been reported to exhibit various acaricidal activities. Previous studies have shown that different types of extractions, such as acetonic, petroleum ether, and ethanolic extracts, display potent repellent activity against adult females and the immature stage of the *T. urticae* mite and its predator *P. persimilis* after 24 and 48 h of treatment [40]. However, there is no existing research specifically investigating the effect of *A. judaica* against ticks, making our study the first to identify its anti-tick activity. Essential oils derived from *A. judaica*, such as piperitone (32.4%), camphor (20.6%), and (*E*)-ethyl cinnamate (8.2%), collected from the Egyptian *Sinai Peninsula*, exhibited repellent activity against *Cowpea weevil* and *Callosobruchus maculatus* insects [41].

In our study, LC–MS analysis of *A. judaica* plant extract identified 74 compounds through molecular networking. Among them, camphor oil exhibited the highest base intensity (Figs 3 and 4, Table 7), suggesting its potential role as a component responsible for the activity of the plant. As camphor oil has been reported to be volatile from different *Artemisia* plant species [42], the environment of humidity and temperature induced in the lab might enhance oil volatile concentration as reported [43]. Therefore, the mechanism by which the extract kills ticks may involve the concentration of camphor oil in the extract, which might affect the ticks inside the incubated Petri dish, leading to mortality that is concentration- and time-dependent (Figs 1 and 3A, Tables 2–5). In conclusion, camphor oil was identified as a major constituent of *A. judaica* plant extract, with multiple reported biological and medicinal uses. However, only a few studies have investigated the efficacy of active essential oil compounds against different tick species and the mechanism underlying their anti-tick activity [32, 44–46].

Several studies have shown the acaricidal activities of different plant oils and extracts [47–49], but they failed to identify the active compounds that might be associated with their activity. Here, we provided some information about the active components that might be associated with the activity of the *A. judaica* extract (the most potent candidate) (Figs 3 and 4, Tables 6 and 7). A few studies have explained the effect of several plant secondary metabolites, such as thymol, carvacrol, 1,8-cineol, n-hexanal, nicotine, dibenzyl-disulfide, and dibenzyl-trisulfide [32, 44–46, 50–52]. These studies focused more on their effect against the tick larval stage than the adults. Meanwhile, most species of ticks that have been studied were *Rhipicephalus microplus*. The only metabolites tested against ticks were thymol, carvacrol, and 1,8-cineol [53].

The cytotoxic potential of the plant extracts used in this study has been evaluated in our previous studies against different cell lines, such as human foreskin fibroblast (HFF) cells [20], human embryonic kidney (293T) cells, and mouse neuroblast (N1E-115) cells [13]. *A. judaica* was reported to be safe against HFF cells with a mean $IC_{50}$ of 316.8 µg/mL [13, 20] and with a mean $IC_{50}$ against 293T and N1E-115 cells of 346.2 and 382.7 µg/mL, respectively [13]. These results support the possible use of this extract in the future studies on tick control in vertebrates' model and the development of acaricidal therapeutics for infected livestock.

## Conclusion

This study's results suggest that *A. judaica* extract and its major component might provide a new acaricidal that is a wild plant extract-based compound and was found to have low cytotoxic potential against different cell lines in our previous studies [13, 20]. Therefore, this extract could potentially control hard ticks in exposed livestock. However, further studies are vital to understanding its mechanism of action before employing it as an acaricidal agent.

## Supporting information

**S1 Table. Latin binomial names of the plants used in this study.** Plants used in this study were collected from the wild survey from the desert roads around Qena Governorate and

Luxor governorate and were identified microscopically in South Valley University herbarium, Faculty of Science, South Valley university, Qena, Egypt. Latin binomial names were also provided in the plant identification letter.
(PDF)

**S1 Fig. A map of wild plant specimens collected from the Egyptian desert.** Map of the plant sample collection sites in Egypt. Plant taxa were collected in a field survey from mid-May 2019 from two sites in the southern part of Egypt in the Qena Governorate desert, and 1 candidate was collected from the Luxor governorate. The collection was performed between 4 AM and 12 PM on 3 successive days. The first site was the Qena-Safaga desert road (X), and the second was the Qena-Sohag desert road (Y). The map was designed by DIVA-GIS 7.5.0 software to illustrate all country roads and governorates; the sample collection sites were approximately determined and are shown in the magnified box as highlighted blue lines on a Google map.
(PDF)

## Acknowledgments

We thank Dr. Mahmoud Abbas Ali (Department of Plant Protection, Faculty of Agriculture, South Valley University, Qena, Egypt) for his excellent technical assistance in the statistical analysis of this manuscript. We are grateful to the great efforts of the South Valley University Workers, South Valley University, Qena, for their help collecting plant specimens from the desert roads. Finally, we thank the deans of the faculties of Veterinary Medicine and Science, South Valley University, Qena, Egypt, for granting us the official permits for plant collection and identification.

## Author Contributions

**Data curation:** Ahmed M. Abdou, Nanang R. Arifeta.

**Formal analysis:** Ahmed M. Abdou, Nanang R. Arifeta.

**Methodology:** Ahmed M. Abdou, Nanang R. Arifeta, Abdel-latif S. Seddek, Samy Abdel-Raouf Fahim Morad, Noha Abdelmageed, Mohamed O. Badry, Rika Umemiya-Shirafuji.

**Project administration:** Yoshifumi Nishikawa.

**Resources:** Abdel-latif S. Seddek, Samy Abdel-Raouf Fahim Morad, Noha Abdelmageed, Mohamed O. Badry, Rika Umemiya-Shirafuji.

**Software:** Ahmed M. Abdou, Nanang R. Arifeta.

**Supervision:** Yoshifumi Nishikawa.

**Validation:** Yoshifumi Nishikawa.

**Visualization:** Ahmed M. Abdou, Nanang R. Arifeta.

**Writing – original draft:** Ahmed M. Abdou.

**Writing – review & editing:** Nanang R. Arifeta, Yoshifumi Nishikawa.

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
