## [Decision Letter · Decision Letter 0]

21 Feb 2024

PONE-D-23-38877Acaricidal activity of Egyptian crude plant extracts against Haemaphysalis longicornis ticksPLOS ONE

Dear Dr. Nishikawa,

Thank you for submitting your manuscript to PLOS ONE. After careful consideration, we feel that it has merit but does not fully meet PLOS ONE’s publication criteria as it currently stands. Therefore, we invite you to submit a revised version of the manuscript that addresses the points raised during the review process.

We look forward to receiving your revised manuscript.

Kind regards,

Brian Stevenson, Ph.D.

Academic Editor

PLOS ONE

Journal Requirements:

This study was supported by KAKENHI Grants from the Japan Society for the Promotion of Science (20F20402 [Y.N.]). 

Reviewers' comments:

Reviewer's Responses to Questions

**Comments to the Author**

1. Is the manuscript technically sound, and do the data support the conclusions?

Reviewer #1: No

Reviewer #2: Partly

2. Has the statistical analysis been performed appropriately and rigorously? 

Reviewer #1: No

Reviewer #2: No

3. Have the authors made all data underlying the findings in their manuscript fully available?

Reviewer #1: Yes

Reviewer #2: Yes

4. Is the manuscript presented in an intelligible fashion and written in standard English?

Reviewer #1: Yes

Reviewer #2: Yes

5. Review Comments to the Author

Reviewer #1: The manuscript on testing plant extracts against the Asian longhorned ticks is quite topical, and important. These extracts were further analyzed on LC-MS. Most of the m/s is straight forward with well written methods, and analysis. However, the reported data is from a non-replicated trial wherein each of the treatment (plant extracts at specific concentrations) were tested on five ticks that were released into a single Petri dish. That would constitute a single replicate.

A few more suggestions:

Mortality criterion can be more refined.

Extraction methods are convincing, however drying them for extended period in open air, and again leaving the applied filter paper for 72 hrs (before testing) is problematic. Some of the compounds (such as camphor etc.,) will evaporate therefore skewing the natural composition of the extracts.

Overall, a nice and much-needed study but unfortunately the above-mentioned shortcomings needed to be addressed for consideration.

Reviewer #2: Major Comments

1. From my reading of the paper it appears that the screen was performed once from a single extraction of each compound. If true, I still think this study remains technically sound as a launching point for future studies given the substantial amount of work that appears to be required to undertake these experiments. However, I would gently request that, if possible, additional replicates be performed with Artemisia judaica extracts to confirm activity. If possible this experiment would be best with a new isolation from a new plant (fresh or dried).

Minor Comments

1. Throughout Text: Change mg/ml to mg/mL

2. Line 50: Citation does not appear to support the claim made. Perhaps citation #3 was meant instead?

3. Line 66: Relevance of the sentence “In a previous study, an in vitro feeding assay system using fipronil and ivermectin was established [10]” is unclear

4. Line 72-73: “Consequently, there is a growing need to explore alternative acaricidal treatments from natural resources, such as plants” The arguments provided do not inherently require alternative treatments be from natural resources. If the authors think natural sources are necessary for a reason not made explicitly clear, they should make that argument. Alternatively, the sentence should be changed to be suggest that one source of alternative acaricidal treatments could be natural resources, such as plants.

5. Line 119: I appreciate that there is a substantial difference between healthy ticks and ticks only moving their appendages, but it’s difficult to justify classifying the latter as dead. If possible, can the data be split between living, moribund, and dead? Or only binary data are recorded, perhaps just using a word slightly less absolute than mortality? If changed, please apply throughout the paper. This is unlikely to change interpretation of the paper so I’ll leave this to author and editor discretion.

6. Line 214: The Solvent and the Total Number of Ticks for Each Plant Concentration are the same for every row, so the columns could be removed and that information added to the table legend.

7. Line 243: What is test is the p-value derived from? What’s the null hypothesis here? Please include in figure legend.

8. “The traditional medicinal uses of A. judaica are treating gastrointestinal disorders, cardiovascular problems, atherosclerosis, arthritis, and skin and eyesight disorders, along with immune system enhancement and anticancer effects [39,40].” Unclear to me whether invoking traditional medicine strengthens the author’s narrative, might suggest removing this sentence.

9. “In the current study, A. judaica showed strong time- and concentration-dependent activities against hard adult female ticks of H. longicornis, suggesting a novel medicinal use of this plant extract as an acaricidal therapeutic.” I would advise caution saying you’ve uncovered a novel “medicinal” role. I think the current study, which did not involve any vertebrate exposures, may go as far to suggest that compounds warrant further study or could be leveraged as an environmental treatment.

10. “Among them, camphor oil exhibited the highest base intensity (Figs. 2 and 3, Table 5), suggesting its potential role as the primary component responsible for the activity of the plant.” I suggest replacing “the primary” with “a”

11. “The mechanism by which the extract kills ticks may involve the concentration of camphor oil in the extract, which might suffocate the ticks inside the incubation Petri dish, leading to mortality that is concentration- and time-dependent (Figs. 1 and 2A, Tables 2 and 5).” I didn’t understand this statement—how would the oil suffocate the ticks?

12. “Therefore, this extract can effectively control hard ticks in exposed livestock.” I would recommend hedging this. “This extract could potentially control”

6. PLOS authors have the option to publish the peer review history of their article (what does this mean?). If published, this will include your full peer review and any attached files.

Reviewer #1: No

Reviewer #2: No

---

## [Author Response · Author response to Decision Letter 0]

30 Jun 2024

Reviewer’s comments PONE-D-23-38877

Journal Requirements:

Answer: Thank you very much for your comment. We have rechecked the format of the manuscript to align it with the journal's style. Changes made to the style, as well as responses to reviewers' comments, were indicated in blue font.

This study was supported by KAKENHI Grants from the Japan Society for the Promotion of Science (20F20402 [Y.N.]). 

Answer: Thank you for your comment. We have updated the funding statement in the cover letter for the manuscript, as required for resubmission. Additionally, we have added the funding section in the revised version of the manuscript, please refer to lines 455-457.

Answer: Thank you for your comment. We have included the revised statement in the cover letter, as required for resubmission.

Answer: Thank you for your comment. Our study did not involve human subjects or acquire any human samples. It is not a retrospective analysis of medical records or archived samples. This information was initially addressed in the Human Participants Research Checklist submitted with the manuscript.

Our research focused on plant samples, and the necessary approvals for collection and identification have been obtained (letters confirming this will be attached during the resubmission process). The relevant statements are included in the manuscript and highlighted in blue font in lines 93-94 and 99-100.

For experiments involving ticks, approval was obtained from the Institutional Animal Care and Use Committee at Obihiro University of Agriculture and Veterinary Medicine (permit numbers 19-74). These statements have been added in the revised version of the manuscript, appearing in line 87

Reviewer #1: 

The manuscript on testing plant extracts against the Asian longhorned ticks is quite topical, and important. These extracts were further analyzed on LC-MS. Most of the m/s is straight forward with well written methods, and analysis. However, the reported data is from a non-replicated trial wherein each of the treatment (plant extracts at specific concentrations) were tested on five ticks that were released into a single Petri dish. That would constitute a single replicate.

Answer: Thank you for your comment. We conducted an additional experiment on the tested extracts, from the most potent candidate (Artemisia judaica), to confirm its efficacy. Please refer to Tables 3 and 5 and Fig 2 for the relevant data.

A few more suggestions:

Mortality criterion can be more refined.

Answer: Thank you very much for your comment. and in response to reviewer 2 comment. In our experimental protocol, we initially defined criteria for mortality, stating that dead ticks or ticks only moving their appendages were to be considered dead. However, it is crucial to note that all data recorded in this study were binary, with tick mortality referring specifically to completely dead ticks, irrespective of the initial protocol statements. We do request keeping the word Mortality as referring to dead ticks in this study.

Extraction methods are convincing, however drying them for extended period in open air, and again leaving the applied filter paper for 72 hrs (before testing) is problematic. Some of the compounds (such as camphor etc.,) will evaporate therefore skewing the natural composition of the extracts.

Answer: Thank you for your comment. In the extraction steps, drying was carried out in a closed environment (closed Petri dish), not in open air, preventing the evaporation of the extract's content. Moreover, the controlled humidity and temperature environment created for the ticks in the laboratory could potentially amplify the concentration of volatile compounds within the incubated Petri dish. This idea resembles what was reported by (Adamo et al., 2022; https://doi.org/10.1038/s41598-022-15164-z).

Reported studies that perform solvent drying in their bioassay were as follows;

https://doi.org/10.4236/ojapps.2022.121005

Mawela, K. G. (2008). The toxicity and repellent properties of plant extracts used in ethnoveterinary medicine to control ticks (Doctoral dissertation, University of Pretoria).

Additionally, according to the LC-MS analysis performed in our study, Camphor was identified as the major component in Artemisia judaica extract (10.38%) among the identified compounds. Consequently, the proposed mechanism of suffocating ticks may be attributed to the volatile activity of camphor oil within the incubator. Please refer to lines 413-415 in the revised version of the manuscript. 

Overall, a nice and much-needed study but unfortunately the above-mentioned shortcomings needed to be addressed for consideration.

Answer: Thank you for your comment. We have incorporated all your suggestions and comments to improve the overall quality of the manuscript.

Reviewer #2: 

Major Comments

1. From my reading of the paper it appears that the screen was performed once from a single extraction of each compound. If true, I still think this study remains technically sound as a launching point for future studies given the substantial amount of work that appears to be required to undertake these experiments. However, I would gently request that, if possible, additional replicates be performed with Artemisia judaica extracts to confirm activity. If possible this experiment would be best with a new isolation from a new plant (fresh or dried).

Answer: Thank you very much for your valuable comment. We agree with your suggestion. To address this concern, we conducted an additional experiment involving the extraction from Artemisia judaica plant extract to confirm its efficacy. However, it's important to note that the isolate used was the same as in our current study. Obtaining new isolates from the original source (Egypt) is currently challenging. Furthermore, we acknowledge the potential impact of variations in climatic conditions, cultivation practices, and the collection of plant materials, and extract production may cause differences in results, as reported by Heimerdinger et al., 2006. Considering these factors, we kindly request your acceptance of the additional experimental data recently conducted as a second replicate, maintaining consistency with the previously tested isolate of Artemisia. We can consider the new isolate in our future studies.

Minor Comments

1. Throughout Text: Change mg/ml to mg/mL

Answer: Thank you very much for your comment We have addressed and corrected the noted issue throughout the revised version of the manuscript as follows;

• Abstract: lines 26,30, and 32.

• Introduction: line 72.

• Material and methods: line 137.

• Tables 1-5.

• Results: lines 191, 196, 202, 212, 213, 220 236,256, 258, 256,262, 281, 283, 288, and 389.

• Discussion: lines 362,364, 367, 370, 434 and 435.

2. Line 50: Citation does not appear to support the claim made. Perhaps citation #3 was meant instead?

Answer: Thank you for your response. We sincerely apologize for this accidental mistake. The correct information has been updated in the relevant reference, and we have rearranged the references’ numbers in both the text and the list in the revised version of the manuscript.

3. Line 66: Relevance of the sentence “In a previous study, an in vitro feeding assay system using fipronil and ivermectin was established [10]” is unclear

Answer: Thank you very much for your comment. We have clarified the information to enhance understanding and clarity. Please refer to lines 69-72 in the revised version of the manuscript, the text was updated as follows: 

“In a previous study, an in vitro feeding assay system using fipronil and ivermectin was established, whereas survival of Ixodes ricinus adult female ticks’ survival was monitored daily over 9 days through a silicone membrane on bovine blood treated with different doses of fipronil and ivermectin ranges from 0.001 to 10 μg/mL [10]”.

4. Line 72-73: “Consequently, there is a growing need to explore alternative acaricidal treatments from natural resources, such as plants” The arguments provided do not inherently require alternative treatments be from natural resources. If the authors think natural sources are necessary for a reason not made explicitly clear, they should make that argument. Alternatively, the sentence should be changed to be suggest that one source of alternative acaricidal treatments could be natural resources, such as plants.

Answer: Thank you very much for your comment. We appreciate your agreement with the correction, the sentence has been revised in the revised version of the manuscript in lines 77-78 as follows;

“One potential source of alternative acaricidal treatments could be natural resources, such as plants. Therefore, this study aimed to evaluate the effectiveness of some crude Egyptian plant extracts against adult female ticks of H. longicornis”.

5. Line 119: I appreciate that there is a substantial difference between healthy ticks and ticks only moving their appendages, but it’s difficult to justify classifying the latter as dead. If possible, can the data be split between living, moribund, and dead? Or only binary data are recorded, perhaps just using a word slightly less absolute than mortality? If changed, please apply throughout the paper. This is unlikely to change interpretation of the paper so I’ll leave this to author and editor discretion.

Answer: Thank you very much for your valuable comment. In our experimental protocol, we initially defined criteria for mortality, stating that dead ticks or ticks only moving their appendages were to be considered dead. However, it is crucial to note that all data recorded in this study were binary, with tick mortality referring specifically to completely dead ticks, irrespective of the initial protocol statements. The determination of dead ticks was meticulously confirmed morphologically and microscopically by at least two authors (A.M.A, N.R.A and R.U-S) in this study. 

To avoid any potential confusion or complication, we have removed the following sentence from the experimental protocol: "Ticks that were paralyzed or moving only their appendages without any walking capability were considered dead." Please see lines 129-130 in the revised version of the manuscript.

6. Line 214: The Solvent and the Total Number of Ticks for Each Plant Concentration are the same for every row, so the columns could be removed and that information added to the table legend.

Answer: Thank you very much for your comment. We have made the necessary adjustments according to your recommendation. The column titles with the data for the Solvent, and the total number of ticks per plant concentration have been deleted, please refer to Table 2 and line 225 in the revised version of the manuscript.

7. Line 243: What is test is the p-value derived from? What’s the null hypothesis here? Please include in figure legend.

Answer: Thank you for your comment. The P-value was derived through the analysis of data and curves generated by the Ldp Line software, rather than from a distinct test. Updated in Tables 4 and 5 foots in lines 297 and 307. This software is specifically designed for calculating probit and regression analysis for this type of data, as outlined by Finney (1971). Please refer to the statistical analysis section in the manuscript methods, specifically in lines 171-178, for further details.

8. “The traditional medicinal uses of A. judaica are treating gastrointestinal disorders, cardiovascular problems, atherosclerosis, arthritis, and skin and eyesight disorders, along with immune system enhancement and anticancer effects [39,40].” Unclear to me whether invoking traditional medicine strengthens the author’s narrative, might suggest removing this sentence.

Answer: Thank you very much for your comment. the sentences along with their corresponding references have been deleted. Please see lines 393-394 in the discussion section and the corresponding references (Khafagy et al., 1988 and Janacković et al., 2015) were deleted in lines 579-583 in the reference section of the revised version of the manuscript.

9. “In the current study, A. judaica showed strong time- and concentration-dependent activities against hard adult female ticks of H. longicornis, suggesting a novel medicinal use of this plant extract as an acaricidal therapeutic.” I would advise caution saying you’ve uncovered a novel “medicinal” role. I think the current study, which did not involve any vertebrate exposures, may go as far to suggest that compounds warrant further study or could be leveraged as an environmental treatment.

Answer: Thank you very much for your comment. We agree that our extract warrants further study in a vertebrate model in future research. In response, we have replaced the following sentence "a novel medicinal use of this plant extract as an acaricidal therapeutic" with the following sentence: "The urgent need to test the efficacy of this plant extract in the treatment of tick infestation in vertebrate models and tick environmental eradication." Please refer to the updated content in lines 398-401 and 436-437 in the revised version of the manuscript.

10. “Among them, camphor oil exhibited the highest base intensity (Figs. 2 and 3, Table 5), suggesting its potential role as the primary component responsible for the activity of the plant.” I suggest replacing “the primary” with “a”

Answer: Thank you very much for your comment. We corrected this grammatical error in line 412 in the revised version of the manuscript as follows; “suggesting its potential role as a component responsible for the activity of the plant”

11. “The mechanism by which the extract kills ticks may involve the concentration of camphor oil in the extract, which might suffocate the ticks inside the incubation Petri dish, leading to mortality that is concentration- and time-dependent (Figs. 1 and 2A, Tables 2 and 5).” I didn’t understand this statement—how would the oil suffocate the ticks?

Answer: Thank you very much for your valuable comment. We change the word “suffocate” to “affect”. Line 416

It was reported that some oils derived from plant extract are known for their volatility (https://doi.org/10.1016/j.microc.2013.07.003). Since camphor has been reported to be a volatile oil derived from different Artemisia plant species (Khayyat and Karimi, 2004; https://doi.org/10.22037/ijps.v1.39440). Moreover, the controlled humidity and temperature environment created for the ticks in the laboratory could potentially amplify the concentration of volatile compounds within the incubated Petri dish. Consequently, the proposed mechanism of suffocating ticks may be attributed to the volatile activity of camphor oil within the incubator. These ideas resemble what was reported by (Adamo et al., 2022; https://doi.org/10.1038/s41598-022-15164-z).

We have provided a comprehensive clarification of the suggested mechanism in

---

## [Editor Report · Decision Letter 1]

3 Jul 2024

Acaricidal activity of Egyptian crude plant extracts against Haemaphysalis longicornis ticks

PONE-D-23-38877R1

Dear Dr. Nishikawa,

We’re pleased to inform you that your manuscript has been judged scientifically suitable for publication and will be formally accepted for publication once it meets all outstanding technical requirements.

Kind regards,

Brian Stevenson, Ph.D.

Academic Editor

PLOS ONE
---

## [Editor Report · Acceptance letter]

12 Jul 2024

PONE-D-23-38877R1 

PLOS ONE

Dear Dr. Nishikawa, 

I'm pleased to inform you that your manuscript has been deemed suitable for publication in PLOS ONE. Congratulations! Your manuscript is now being handed over to our production team.

Kind regards, 

on behalf of

Prof. Brian Stevenson 

Academic Editor

PLOS ONE